# Comparison of regulatory approval system for medicines in emergency among Japan, the United States, the United Kingdom, Europe, and China

Kazuki Edo[1], Masahide Kawano[2,3]*, Hideki Maeda[1,2]*

1 Department of Regulatory Science, Faculty of Pharmacy, Meiji Pharmaceutical University, Tokyo, Japan,
2 Department of Regulatory Science, Graduate School of Pharmaceutical Science, Meiji Pharmaceutical University, Tokyo, Japan, 3 Department of Medical Affairs Japan, Astellas Pharma Inc., Tokyo, Japan

* maeda@my-pharm.ac.jp (HM); d226752@std.my-pharm.ac.jp (MK)

## Abstract

The approval of pharmaceuticals in response to the COVID-19 pandemic is a global concern, and during emergencies, emergency approval or authorization systems that enable the rapid use of unapproved drugs to maintain national health are essential. However, there is limited research comparing these systems across countries and their effects. This cross-sectional study examined such systems in Japan (JP), Europe (EU), the United Kingdom (UK), and China (CN) for pharmaceuticals (n = 23) authorized under Emergency Use Authorization (EUA) in the United States (US) between December 2019 to July 2023. As of the end of July 2023, JP had granted approval or permission for 14 drugs (60.9%), EU for 14 (60.9%), UK for 12 (52.2%), and CN for three (13.0%). An examination of the developmental status of the 23 drugs revealed that JP had 6 drugs (26.1%), the EU had 3 drugs (8.7%), the UK had 5 drugs (21.7%), and CN had 16 drugs (69.6%) yet to be developed. The US had significantly more granted permissions and developed drugs, while CN the least. The UK had a significantly shorter period for approval than the US and the shortest overall. The EU had the shortest period from the issuance of EUA to approval dates. Although not statistically significant, JP had the longest duration until unapproved drugs could be used. Pharmaceuticals granted usage permission under the EUA in JP, the EU, and the UK were developed or on the market in over 70% of cases, whereas in CN, more than two-thirds were yet to be developed. This suggests that CN may not actively utilize pharmaceuticals from other countries for COVID-19 treatment and may rely on its own. When comparing the emergency approval and permission systems of each country, the most significant difference was in the type of system granting approval.

**Data Availability Statement:** In Japan (JP), information was collected from the Pharmaceuticals and Medical Drugs Directory (PMDA), the Ministry of Health, Labour and

Welfare (MHLW) website, and MHLW press releases. (https://www.pmda.go.jp/PmdaSearch/iyakuSearch, https://www.mhlw.go.jp/index.html) In the United States (US): Information was collected from Drugs@FDA, the Emergency Use Authorization (EUA) page of the FDA web site, and FDA press releases. (https://www.fda.gov/emergency-preparedness-and-response/mcm-legal-regulatory-and-policy-framework/emergency-use-authorization) Europe (EU): Information was collected from the Conditional Marketing Authorization (CMA) page of the EMA web site and EMA press release information. (https://www.ema.europa.eu/en/glossary-terms/conditional-marketing-authorisation) In United Kingdom (UK): Information was collected from MHRA Products Substance Index, MHRA web site page on Temporary Use authorization, and MHRA press release information. (https://www.gov.uk/government/publications/freedom-of-information-responses-from-the-mhra-week-commencing-14-march-2022/freedom-of-information-request-on-the-different-legal-classifications-for-vaccines-foi-21415) CHINA (CN): Information was collected from the official NMPA website, CENTER FOR DRUG EVALUATION, NMPA. (https://english.nmpa.gov.cn/index.html, https://english.nmpa.gov.cn/2019-07/19/c_389169.htm)

**Funding:** The author(s) received no specific funding for this work.

**Competing interests:** M.K. is an employee of Astellas Pharma Inc. Other authors have no conflicts of interest to declare. This does not alter our adherence to PLOS ONE policies on the sharing of data and materials.

## Introduction

The approval of pharmaceuticals during the Coronavirus Disease 2019 (COVID-19) pandemic has been a global concern [1–3]. In emergencies, such as the COVID-19 pandemic, a system that allows the use of unapproved drugs is essential for maintaining the health and public hygiene of the population. Following the onset of the COVID-19 pandemic, pharmaceutical companies worldwide have rapidly developed treatments and vaccines with governments strongly supporting their practical applications [4]. From a regulatory perspective, under normal circumstances, drug approval typically requires approximately one year from application to approval in many countries, including Japan, the United States, and European countries in the European Union. However, for pharmaceuticals addressing COVID-19, many countries have prioritized and expedited reviews. Emergency regulatory frameworks have been utilized, leading to review periods ranging from days to months [5]. In the United States (US), the Emergency Use Authorization (EUA) system [6] was employed to facilitate the early practical application of pharmaceuticals.

The EUA of US differs from the regular drug approval system in that it conditions approval on the statement "the product may be effective" regarding the effectiveness and on demonstrating that "the known and potential benefits of the product outweigh the known and potential risks." Unlike the standard drug approval process, the EUA does not require the submission of materials in the Common Technical Document (CTD) format, and in some cases, there may not be a Good Manufacturing Practice (GMP) inspection of quality aspects. The relatively simplified documentation required for EUA applications is believed to ease the burden on pharmaceutical companies developing drugs globally. This, in turn, contributes to incentives for prioritizing development considering the reduced application workload. Furthermore, after obtaining an Emergency Use Authorization (EUA), there are cases in which a re-evaluation is conducted based on post-market data for a certain period. In some instances, drugs may obtain full approval through this re-evaluation, whereas in cases where effectiveness is not demonstrated and there are safety concerns, withdrawal might occur [6]. In Europe (EU) [7], the United Kingdom (UK) [8], and China (CN) [9], early approval and permission systems have been established as frameworks for using unapproved drugs during emergencies [10]. In the EU, conditional marketing authorization is valid for one year and renewable annually [11]. In the UK temporary use authorization is valid for one year and can be transferred to a normal authorization. CN has conditional market approval and emergency use of vaccines [12].

In Japan (JP), there has been a special approval system as an emergency regulatory framework during the COVID-19 pandemic [13]. However, because drugs developed in JP are not covered by this system [5], a new system called emergency approval has been in effect since May 2022 [14].

Systems allowing the use of unapproved pharmaceuticals during emergencies have been individually established in each country, and there is limited research that compares and clarifies the differences between these national systems. Therefore, this study aimed to compare the emergency approval or usage systems for unapproved pharmaceuticals in JP, the US, the EU, the UK, and CN with reference to the U.S., focusing on investigating the approval and usage permission systems during emergencies such as pandemics, and aimed to identify the commonalities and differences in each system. European countries under the European Union were considered as one country for the purposes of this study. This study also explored how pharmaceuticals granted usage permission under the US EUA were approved or permitted in other countries (JP, EU, UK, and CN) and conducted a comparative analysis of the emergency approval or usage permission systems in each country. Finally, the study examined whether

the differences in the systems for unapproved pharmaceuticals during emergencies in different countries affected drug access.

## Materials and methods

### 1. Data collection

Information on pharmaceuticals granted usage permission under the US Emergency Use Authorization (EUA) and situations in other countries (JP, EU, UK, CN) between December 1, 2019, and July 31, 2023, during the COVID-19 pandemic, was obtained from public sources [15–19] to create a proprietary database. The specific investigation parameters included approval or usage permission status, approval or usage permission dates, JP approval application dates, expiration dates of approval or usage permission, review periods, the duration from EUA issuance to approval or usage permission date in each country, and the developmental stage. Data collection was conducted through the official websites of the regulatory authorities in each country [15–19] and ClinicalTrials.gov. Each website was reviewed, and the data were manually extracted and verified by a researcher. Data on the status of approval/use authorization, date of approval/use authorization, date of JP approval, date of withdrawal of approval/use authorization, review period, and period from the date of EUA issuance to the date of approval/use authorization in each country were collected from the regulatory agency web sites in each country as shown below. We also collected data from ClinicalTrials.gov on the pre-market phase of the drug, the most recent data available as of July 31, 2023. Duplicates were eliminated in the collected data, formatted uniformly, and compiled into a database.

In the United States (US): Information was collected from Drugs@FDA, the Emergency Use Authorization (EUA) page of the FDA web site, and FDA press releases.

(https://www.fda.gov/emergency-preparedness-and-response/mcm-legal-regulatory-and-policy-framework/emergency-use-authorization)

In Japan (JP), information was collected from the Pharmaceuticals and Medical Drugs Directory (PMDA), the Ministry of Health, Labour and Welfare (MHLW) website, and MHLW press releases.

(https://www.pmda.go.jp/PmdaSearch/iyakuSearch, https://www.mhlw.go.jp/index.html)

Europe (EU): Information was collected from the Conditional Marketing Authorization (CMA) page of the EMA web site and EMA press release information.

(https://www.ema.europa.eu/en/glossary-terms/conditional-marketing-authorisation)

In United Kingdom (UK): Information was collected from MHRA Products Substance Index, MHRA web site page on Temporary Use authorization, and MHRA press release information.

(https://www.gov.uk/government/publications/freedom-of-information-responses-from-the-mhra-week-commencing-14-march-2022/freedom-of-information-request-on-the-different-legal-classifications-for-vaccines-foi-21415)

CHINA (CN): Information was collected from the official NMPA website, CENTER FOR DRUG EVALUATION, NMPA.

(https://english.nmpa.gov.cn/index.html, https://english.nmpa.gov.cn/2019-07/19/c_389169.htm)

### 2. Content of investigation

The following two investigations were conducted during this study.

A.  Identified pharmaceuticals were granted usage permission under US Emergency Use Authorization (EUA) from December 1, 2019, to July 31, 2023, and the processes through

which these identified pharmaceuticals were approved or permitted in each country were investigated and compared. The specific items studied are (1) and (2).

1. Review periods and the duration from the issuance date of Emergency Use Authorization (EUA) to the approval date in each country

2. Presence or absence of approval/usage permission for each pharmaceutical in each country and developmental stage

B. We identified emergency approval/usage permission systems in each country (the US, JP, EU, UK, and CH) and conducted a comparison of these systems. The specific items studied were (1) and (2) as of July 31, 2023.

1. System used for approval or permission.

2. The presence or absence of legislative amendments or new enactments regarding approval/usage permission systems during the pandemic, along with the types and timing of such changes.

This study was conducted in accordance with the Strengthening the Reporting of Observational Studies in Epidemiology (STROBE) reporting guidelines [20] for cross-sectional studies. Kazuki Edo (KE) and Masahide Kawano (MK) independently conducted the investigation parameters. Any questions or issues were discussed between the two authors and were adjusted when necessary. Any disagreement was resolved by a third reviewer [Hideki Maeda (HM)].

## 3. Statistical methods

Microsoft® Excel and R-4.4.1 were used for statistical analysis. The Mann-Whitney U test was used for the review period and the period from the date of EUA issuance to the date of approval in each country, and the Fishers exact test was used for the development stage of each drug in each country. The statistical significance level was set at 5% (two-tailed). Bonferroni correction was applied to the Mann-Whitney U test to account for multiplicity. Regarding the temporal analysis, because our main objective was to make a US-based comparison with respect to each country's urgent approval system, we tracked key time points such as approval dates and withdrawal dates within the study period. For drugs that were submitted to the authorities but not approved during the study period, their status was recorded and considered in the overall analysis. While a detailed analysis with time as a factor, such as a survival analysis, would be beneficial, the main objective of this study was to compare regulatory systems in different countries. Therefore, a descriptive and comparative approach was adopted in this study.

## Results

### 1. Breakdown of EUA pharmaceuticals issued in response to COVID-19

During the COVID-19 pandemic period (December 1, 2019, to July 31, 2023), 23 pharmaceuticals were granted usage permission under the US EUA. After re-evaluation by the FDA, five drugs (21.7%) received full approval, 13 drugs (56.5%) had their EUA usage permission continued, and five drugs (21.7%) had their EUA revoked or discontinued (Fig 1). "EUA usage permission continued" is defined as a drug in the US as of July 31, 2023, that has not yet been granted an EUA and has not moved to full approval. This definition includes all drugs that have been granted an EUA and remain in EUA status without having reached full approval by the specified date.

As of the end of July 2023, an investigation into the approval/usage permission status of medicinal products that had been previously licensed for use under the EUA in the period

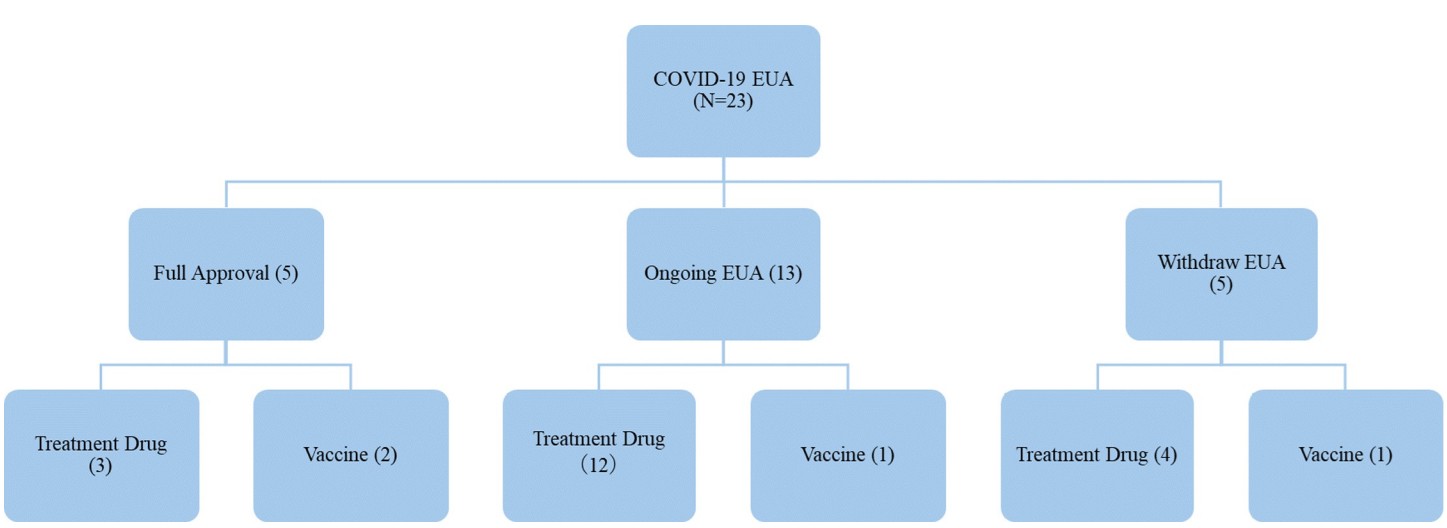

**Fig 1. Flowchart of drugs authorized for use under Emergency Use Authorization by FDA against COVID-19.** COVID-19: Coronavirus disease 2019, EUA: Emergency Use Authorization.

under study as of the end of July 2023 revealed that JP had obtained approval or permission for 14 drugs (60.9%), the EU for 14 drugs (60.9%), the UK for 12 drugs (52.2%), and CN for 3 drugs (13.0%). In comparison with each country, the US had a significantly highest number of usage permissions granted (P = 0.0002, χ2 test) (Table 1). Additionally, an investigation into the developmental status of pharmaceuticals granted usage permission under the EUA as of the end of July 2023 in each country showed that the EU had three drugs (8.7%), the UK had five drugs (21.7%), JP had six drugs (26.1%), and CN had 16 drugs (69.6%) that were yet to be developed. The proportion of undeveloped pharmaceuticals in CN was significantly larger compared to other countries (P<0.0001, Fishers exact test) (Table 2). This indicates that the development status of drugs is not uniform across countries. In particular, CN has a significantly higher percentage of undeveloped drugs than other countries, suggesting that there may be differences in drug development processes and priorities.

## 2. Review period in each country of pharmaceuticals given usage permission under EUA

At the end of July 2023, an investigation into the review periods in each country for pharmaceuticals granted usage permission under US Emergency Use Authorization (EUA) was conducted. The median (Interquartile Range; IQR) review periods were as follows: US 64.5 (35.0–104.3) days, JP 37.5 (21.0–91.3) days, EU 33 (21.0–114.5) days, and UK 14 (6.3–59.8) days. The shortest review period was observed in the UK, where the mean was 50.5 days earlier (Fig 2). There was a statistically significant difference in the review period between US-UK

**Table 1. Status of approval/use authorization in each country for drugs granted the Emergency Use Authorization by FDA.**

|  | US | JP | EU | UK | CN | Total | *P value*\* |
|---|---|---|---|---|---|---|---|
| Approval or Use Authorization | 18 (78.3%) | 14 (60.9%) | 14 (60.9%) | 12 (52.2%) | 3 (13.0%) | 61 (53.0%) | P = 0.0001 |
| Withdrawal or Undeveloped | 5 (21.7%) | 9 (39.1%) | 9 (39.1%) | 11 (47.2%) | 20 (87.0%) | 54 (47.0%) |  |
| Total | 23 (100%) | 23 (100%) | 23 (100%) | 23 (100%) | 23 (100%) | 115 (100%) |  |

\*: chi-square test, US: the United States, JP: Japan, EU: Europe, UK: the United Kingdom, CN: China

**Table 2. Development status in each country for drugs granted the Emergency Use Authorization by FDA.**

|  | US | JP | EU | UK | CN | Total | *P value** |
|---|---|---|---|---|---|---|---|
| Approval | 18 (78.3%) | 14 (60.9%) | 14 (60.9%) | 12 (52.2%) | 3 (13.0%) | 61 (53.0%) | *P*<0.0001 |
| Phase 1 or Phase 2 | 0 (0%) | 0 (0%) | 0 (0%) | 0 (0%) | 2 (8.7%) | 2 (1.7%) |  |
| Phase 3 | 0 (0%) | 3 (13.0%) | 4 (17.4%) | 6 (26.1%) | 2 (8.7%) | 15 (13.0%) |  |
| Under application | 0 (0%) | 0 (0%) | 1 (4.4%) | 0 (0%) | 0 (0%) | 1 (0.9%) |  |
| Withdraw | 5 (21.7%) | 0 (0%) | 2 (8.7%) | 0 (0%) | 0 (0%) | 7 (6.1%) |  |
| Undeveloped | 0 (0%) | 6 (26.1%) | 2 (8.7%) | 5 (21.7%) | 16 (69.6%) | 29 (25.2%) |  |
| Total | 23 (100.0%) | 23 (100.0%) | 23 (100.0%) | 23 (100.0%) | 23 (100.0%) | 115 (100.0%) |  |

*: chi-square test, US: the United States, JP: Japan, EU: Europe, UK: the United Kingdom, CN: China

(P = 0.0037), JP-UK (P = 0.0391). No statistically significant difference in examination duration was found between US-JP (P = 0.9553), US-EU (P = 0.4022), JP-EU (P = 4.0866), and EU-UK (P = 0.1509). As for the data on CN, no comparison was made because the application date could not be identified based on publicly available information from regulatory authorities.

## 3. Duration from the date of usage permission granted under the EUA to the approval/usage permission dates in each country

By the end of July 2023, an investigation was conducted on the duration from the date of usage permission granted under the EUA to the approval or usage permission dates in each country for pharmaceuticals for which the EUA continued. The duration until approval or usage permission was, in median order, shortest in the EU, followed by the UK and JP (Fig 3). The median durations were 37 (-97.5–270.5) days for EU, 54 (-39.3–167.5) days for UK, and 139.5 (-37.8–231.3) days for JP. Although no statistically significant differences were observed, JP

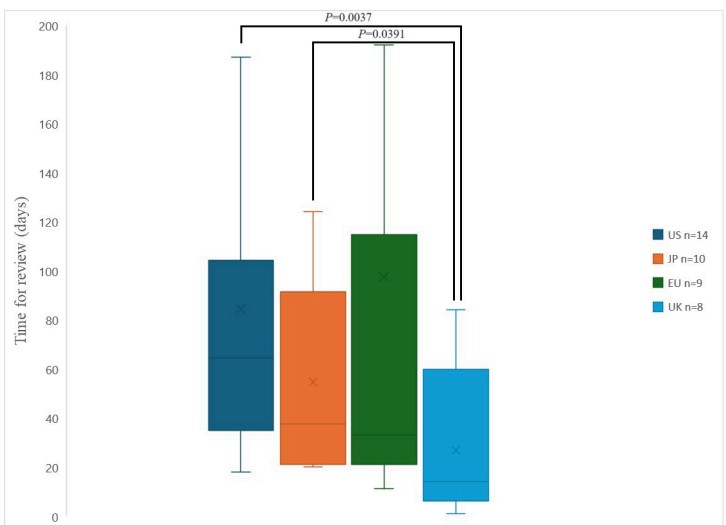

**Fig 2. Review period of regulatory authority in each country for drugs authorized for use under Emergency Use Authorization by FDA against COVID-19.** The horizontal line in each box represents the median. The x mark represents the average values. The lines at the top and bottom edges of each box represent the 75th and 25th percentiles, respectively. The upper and lower limits of the vertical lines represent the maximum and minimum values, respectively. US: United States; JP: Japan; EU: Europe; UK: United Kingdom.

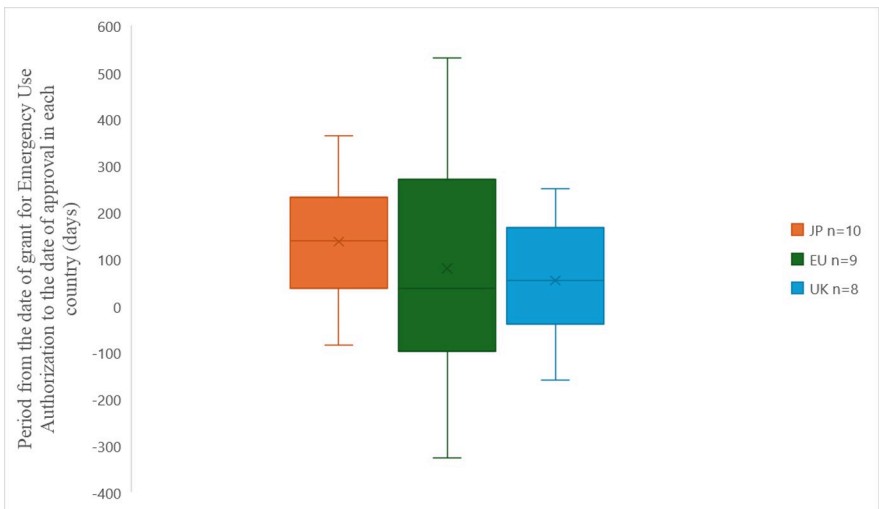

**Fig 3. Period from the date of grant for Emergency Use Authorization to the date of approval in each country.**
The horizontal line in each box represents the median. The x mark represents the average values. The lines at the top and bottom edges of each box represent the 75th and 25th percentiles, respectively. The upper and lower limits of the vertical lines represent the maximum and minimum values, respectively. JP: Japan, EU: Europe, UK: the United Kingdom.

had the longest duration from EUA usage permission to approval, with a median of approximately 140 days before approval. As there are only three approved pharmaceuticals in CN, we did not compare this data.

The duration from the date of usage permission granted under the EUA to the approval/usage permission was not a statistically significant difference for all between JP-EU (P = 0.9127), JP-UK (P = 0.3829), and EU-UK (P = 1.7006).

## 4. System for approval/usage permission under emergency circumstances in each country

An overview of the investigation of emergency approval/usage permission systems in each country is presented in Table 3. Drugs approved with expedited programs (EU: Conditional Marketing Authorisations, JP: Special Approval for Emergency, CN: Conditional Market Approval) can be used even after an emergency situation has passed. However, use authorization (US: Emergency Use Authorization, UK: Temporary Use Authorisation, CN: Chinese Vaccine Management Law) is terminated after an emergency situation has passed, and the drug can no longer be used. The most significant difference in each country's system was whether it operated under an approval or a usage permission framework. Approval refers to the process by which a drug meets full regulatory requirements and is authorized for permanent use. Use authorization is the process by which a drug is allowed to be used temporarily when a rapid response is required, such as in an emergency, and is usually based on limited data. The added data may trigger a move to ordinary full approval or withdrawal of permission to use. The US and UK employed a usage permission system, whereas JP, the EU, and CN operated under an approval system. Additionally, in JP, the UK, and CN, new systems were established in response to the COVID-19 pandemic. In JP, an emergency approval system has been introduced, allowing the approval of pharmaceuticals not approved overseas. In JP, a 'Special Approval System' initially existed as an emergency approval system for drugs. This system aims to expedite the regular review process and approve products for market use in cases

**Table 3. Summary of emergency approval/use authorization regulations in each country.**

|  | JP | US | EU | UK | CN |
|---|---|---|---|---|---|
| Regulation name | Special Approval for Emergency (SAFE) | EUA(Emergency Use Authorization) | Conditional marketing authorisation | Temporary use authorisation | conditional market approval |
| Established year | 2010 | 2004 | 2006 | 2020/10/17(174A inserted) | 2019/12/1 |
| Approval or Use Authorization | Approval | Use authorisation | Conditional marketing authorisations | Use authorisation | conditional market approval |
| Target | Drugs/Medical device/ Biological products | Drugs/Medical device/Biological products | Drugs/Biological products | medical product | Drugs/Medical device/ Biological products |
| Whether there are any amendments or new establishments in the law | Yes | No | No | Yes | Yes |
| Target laws and regulations | Emergency approval | - | - | Conditional Marketing Authorisation (CMA) | Chinese vaccine management law |
| Content | It is now possible to approve a drug even if it is not already approved overseas. | - | - | Introducing the Conditional Marketing Authorization (CMA) system | Allow emergency authorization use of vaccines |

US: the United States, JP: Japan, EU: Europe, UK: the United Kingdom, CN: China

of urgency or medical necessity. As the Special Approval System primarily focuses on approving overseas drugs, a new emergency approval system capable of approving JP-origin pharmaceuticals was established in 2023 [14]. In the UK, a conditional marketing authorization system was implemented. Additionally, the Medicines and Healthcare Products Regulatory Agency (MHRA) established Temporary Use Authorization in 2020. This authorization simplifies the regular review process, enabling rapid commercialization through use authorization [8]. The CN introduced a Vaccine Administration Law enabling the emergency use of vaccines and implemented conditional market approval for CN in 2019 [9]. This system expedites the progress of the regular review process to meet urgent medical needs, allowing early patient access to pharmaceuticals.

## Discussion

COVID-19, a global pandemic caused by the novel coronavirus, has led to widespread infections and severe health crises. The development of drugs for the treatment of COVID-19 has become a global challenge. While methods for rapid approval and usage permissions vary globally, common practices include provisional approval, expedited approval, or expedited reviews through emergency use authorization. A significant difference in systems across countries lies in the choice between approval and authorization, as shown in Table 3. In JP, a" Special Approval System" initially existed as an emergency approval system for drugs. This system aims to expedite the regular review process and approve products for market use in cases of urgency or medical necessity. Specifically, the system was established in May 2010 in response to the novel influenza A (H1N1) outbreak and has been applied to the COVID-19 vaccine [5]. As the Special Approval System primarily focuses on approving overseas drugs, a new emergency approval system capable of approving JP-origin pharmaceuticals was established in 2023 [14]. As of November 2022, only one drug, ensitrelvir, has been approved by the emergency approval system in Japan [21]. This emergency approval system is expected to serve as Japan's early approval mechanism during this pandemic. In the US, the EUA has existed since 2004 [6]. It is designed to expedite regular review processes and address threats to public health

emergencies, allowing patients first access to new pharmaceuticals. The EUA is not an approval system but rather a permission for emergency use, and it has been utilized for COVID-19-related pharmaceuticals and vaccines. After obtaining the EUA, some drugs undergo re-evaluation based on postmarketing data for a certain period. In some cases, full approval may be obtained, whereas in cases where effectiveness is not demonstrated and safety concerns arise, the drug may be withdrawn [6]. In the EU, the European Medicines Agency (EMA) oversees the Conditional Marketing Authorization (CMA) [7]. This approval system allows patients to access pharmaceuticals more quickly than usual when there is a significant medical need, and its effectiveness and safety can be confirmed. In the UK, the Medicines and Healthcare Products Regulatory Agency (MHRA) established Temporary Use Authorization in 2020. This authorization simplifies the regular review process, enabling rapid commercialization through use authorization [8]. The National Medical Products Administration (NMPA) implemented conditional market approval for CN in 2019 [9]. This system facilitates expedited progress of the regular review process to meet urgent medical needs, allowing early patient access to pharmaceuticals. Thus, new systems and amendments have been introduced worldwide to facilitate expedited reviews for first access to COVID-19 pharmaceuticals.

In this study, considering the frequent approval of drugs in the US and the tendency of the US to obtain approvals quickly [22], we used pharmaceuticals approved by the US EUA as a control and examined the situation in various countries. Specifically, we investigated the status of pharmaceuticals authorized by the US EUA for use from December 1, 2019, to July 31, 2023, in JP, the EU, the UK, and CN. As a result, more than 70% of the pharmaceuticals authorized by the EUA for COVID-19 had developed or were in the market in JP, the EU, and the UK. In contrast, CN has over two-thirds of its pharmaceuticals yet to be developed, suggesting a distinct approach from other countries. Therefore, CN may have independently developed its own pharmaceuticals for the COVID-19 pandemic. Furthermore, in terms of pharmaceuticals for COVID-19, 19 of 23 pharmaceuticals authorized by the EUA had received authorization in the US, surpassing JP, the EU, the UK, and CN. Additionally, an analysis of EUA pharmaceutical review periods revealed that the UK had the shortest period. This was likely facilitated by the introduction of a usage authorization system through UK legal amendments (insertion of 174A into Regulation 174 of the Human Medicine Regulations 2012; HMR2012). Indeed, an examination of pharmaceuticals before and after the implementation of the usage authorization system in the UK showed that pharmaceuticals authorized after the implementation had shorter review periods than those authorized before. Regarding the period from the EUA issuance date to the approval date in each country for pharmaceuticals authorized by the EUA, there was no significant difference between the countries examined. However, approval was relatively delayed in Japan (4.7 months), raising the possibility of drug lag or drug loss in the use of unapproved drugs during emergencies [23, 24]. An emergency approval/use authorization system should aim to expedite access to superior unapproved drugs in emergencies to promptly prevent the escalation of health risks. We believe that such a system must be effective in every country. In emergency situations, an expedited approval process is required. This includes special procedures that simplify and expedite the ordinary approval process, for example, the EUA in the U.S. and the CMA in the EU. Such schemes are essential to enable rapid access to medicines in emergency situations and to respond to crises such as pandemics. It is also important to strike a balance between safety and efficacy. The emergency approval process must ensure that known and potential risks are minimal, even when approvals are based on limited data [25]. This ensures that treatments are safe and effective for patients, even in emergency situations. Additionally, flexible data requirements are necessary. In emergency situations, approvals must be based on limited data without waiting for complete clinical data [25]. This includes flexible data requirements that simplify the documentation required for

applications and allow for expedited review. Additionally, transparency is critical. The emergency approval process requires that authorities set clear criteria for the approval process and decisions and provide appropriate information to healthcare professionals and the public [26]. This ensures that reliable information is shared, and effective treatment is provided during emergencies. Finally, pharmacovigilance of emergency-approved drugs is also important. Pharmacovigilance is executed and, if necessary, the approval is reviewed or withdrawn to continuously assess safety and efficacy.

A regulatory framework with the above features will function effectively in emergency situations, enabling the prompt and safe delivery of medicines.

This study has several limitations. First, it focuses on drugs that were granted the EUA; therefore, it does not consider drugs that were not subject to the EUA, but were first approved or granted authorization for use in countries outside the US. Second, the study only addressed pharmaceuticals and did not include medical devices or diagnostic drugs. Third, the countries included in the study were limited to the US and four other countries, excluding other developed or developing nations, which may introduce some variation in global trends. Fourth, as a result of the limitations mentioned, there may be potential bias in the results.

## Conclusions

This study aimed to conduct an international comparison of emergency approval/use authorization systems for pharmaceuticals during emergencies in Japan, the United States, Europe, and China (JP, US, EU, UK, and CN), focusing on 23 pharmaceuticals authorized for use against COVID-19 through the EUA granted by the FDA. While over 70% of the EUA-approved drugs have been developed or marketed in the JP, EU, and UK, more than 2/3 of them are yet to be developed for CN. This suggests that CN may not actively utilize pharmaceuticals from other countries for COVID-19 and instead rely on their own pharmaceuticals. JP had the longest duration from EUA issuance to approval among the countries studied. When comparing the emergency approval and permission systems of each country, the most significant difference observed was in the type of the systems granting approval. We aim to conduct further research on emergency approval/use authorization systems for pharmaceuticals in the future.

## Supporting information

**S1 Checklist. STROBE statement—checklist of items that should be included in reports of *cross-sectional studies*.**
(DOCX)

## Author Contributions

**Conceptualization:** Kazuki Edo, Hideki Maeda.

**Data curation:** Kazuki Edo.

**Formal analysis:** Kazuki Edo, Masahide Kawano, Hideki Maeda.

**Investigation:** Kazuki Edo, Masahide Kawano, Hideki Maeda.

**Methodology:** Kazuki Edo, Masahide Kawano, Hideki Maeda.

**Resources:** Kazuki Edo.

**Software:** Kazuki Edo.

**Supervision:** Hideki Maeda.

**Writing – original draft:** Kazuki Edo, Masahide Kawano, Hideki Maeda.

**Writing – review & editing:** Masahide Kawano, Hideki Maeda.

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
