## [Decision Letter · Decision Letter 0]

11 Jul 2024

PONE-D-24-08507Comparison of regulatory approval system for medicines in emergency among Japan, the United States, the United Kingdom, Europe, and China.PLOS ONE

Dear Dr. KAWANO,

Thank you for submitting your manuscript to PLOS ONE. After careful consideration, we feel that it has merit but does not fully meet PLOS ONE’s publication criteria as it currently stands. Therefore, we invite you to submit a revised version of the manuscript that addresses the points raised during the review process.

We look forward to receiving your revised manuscript.

Kind regards,

Ashraful Hoque, MD

Academic Editor

PLOS ONE

Journal Requirements:

"This study was partially funded by a grant from Research on Regulatory Science of Pharmaceuticals and Medical Devices, Health, Labour and Welfare Policy Research (21KC2006)"

 "This study was partially funded by a grant from Research on Regulatory Science of Pharmaceuticals and Medical Devices, Health, Labour and Welfare Policy Research (21KC2006)"

"M.K is the employee of Astellas Pharma Inc."

7. Please upload a new copy of Figure xxxx as the detail is not clear. Please follow the link for more information: https://blogs.plos.org/plos/2019/06/looking-good-tips-for-creating-your-plos-figures-graphics/" https://blogs.plos.org/plos/2019/06/looking-good-tips-for-creating-your-plos-figures-graphics/"

**Additional Editor Comments:**

Please provide a comprehensive explanation of any comments mentioned by Reviewer 1, if applicable, and make the necessary modifications to the revised version.

Reviewers' comments:

Reviewer's Responses to Questions

**Comments to the Author**

1. Is the manuscript technically sound, and do the data support the conclusions?

Reviewer #1: Partly

Reviewer #2: Partly

2. Has the statistical analysis been performed appropriately and rigorously? 

Reviewer #1: No

Reviewer #2: No

3. Have the authors made all data underlying the findings in their manuscript fully available?

Reviewer #1: No

Reviewer #2: Yes

4. Is the manuscript presented in an intelligible fashion and written in standard English?

Reviewer #1: Yes

Reviewer #2: Yes

5. Review Comments to the Author

Reviewer #1: The article "Comparison of regulatory approval system for medicines in emergency among Japan, the United States, the United Kingdom, Europe, and China" addresses an important and timely issue with respect to the variability in emergency drug regulatory approval process for various countries, focusing on pharmaceuticals related to COVID-19.

Unfortunately, in its current form I cannot recommend this article for publication. There are several significant concerns regarding elements of study design and statistical evaluation.

Details can be found in the respective sections to follow:

Introduction:

The authors provide a strong motivation for the work, highlighting the heterogeneity in the drug review processes across countries.

- More detail could be provided around the specific variances across each of the countries processes that will be discussed.

- There is also an overall lack of related work exploring variance between countries in related topics such as COVID-19 vaccine approval.

e.g.

Lythgoe, Mark P., and Paul Middleton. "Comparison of COVID-19 vaccine approvals at the US Food and Drug administration, European Medicines agency, and health Canada." JAMA Network Open 4.6 (2021): e2114531-e2114531.

Materials and Methods:

Data collection

- This section needs expansion. The "proprietary dataset" is not sufficiently detailed. While citations for the data sources are valuable, the authors note that data collection was "primarily conducted" through official webpages. How was the remainder of data collected?

- More detail is needed regarding data extraction. Where in each of the webpages were data elements extracted?

- From the links provided, was any criterion or processing applied to identified data elements to create the final dataset.

- The use of a single reviewer is concerning. Given manual data extraction, dual review should have been performed to confirm quality of data elements.

Content of investigation

- The hypothesis here does not seem to match the objectives outlined in the title/introduction. The article is positioned as a comparison of the approval of medications between multiple counties, however (A) clarifies this is restricted to only the subset of drugs approved by the United States EUA (and then comparing their approval in other countries). A full comparison between approval mechanisms would include the potential for drugs approved in other countries and not through the EUA. This is mentioned in 1 sentence for limitations, but this is not sufficient. Revision to the framing of the manuscript is needed.

--- This is also somewhat at odds with later analysis for timeline for EUA, in which some countries had negative timelines, suggesting their use of drug before FDA approval. The bi-directional framing of the analysis is needed.

- The study does not appear to follow the complete STROBE guidelines. Please attach the checklist document with associated page numbers

Statistical Methods

I have several concerns regarding the selection of statistical tests - discussed in the results section below.

Results:

It is unclear what had their "EUA permission continued" means, is this only those drugs continued with the July 2023 timeframe? Or continued at any point during this period after EUA. The timeframe is ambiguously defined, more information is needed.

Analysis 1 (percentage of drugs approved by country - i.e. Table 1) appears flawed.

- First, drugs that are not approved in other countries, are listed as "withdrawal", however is it not possible these drugs were not even considered for emergency approval? The breakdown of only approved/withdraw seems problematic. Please expand/explain.

- The United States is included in the analysis. All other countries must consider equal to, or fewer total drugs as the superset of drugs is the 23 total drugs that obtained FDA approval. As we are considering non-approved drugs to be withdrawn it seems odd to include the United States in which all 23 drugs were approved with some withdrawn.

- This issue appears again in Table 2, which undeveloped drugs cannot exist for the US, yet are considered in the table.

- Moreover Chi-squared is not an appropriate test for this data. The number of 0-cells is high. Fishers exact should have been performed.

Review period analysis.

- Use of t-tests is somewhat odd. The authors report median/IRQ indicating awareness of a non-normal distribution (heavy tailed). As such, I would expect to see non-parametric Mann-U or similar tests used.

- Figure 2 provides no information as to why only a subset of drugs were considered from each country (i.e. 14 of 23 drugs were considered for the US). I expect perhaps that the review period was not available for all drugs, but if that is the case, more information is needed to discuss exclusions, and post-hoc analysis is needed for potential bias due to latent factors related to lack of reporting (i.e. quick withdrawal).

- I expect pairwise t-tests were used to obtain the results between countries. However (1) results are not reported for "non-significant" countries. However, this information must be made available by text or within the figure.

- No correction for multiple comparisons was used and must be added to account for the repeated pairwise testing.

- Again, pairwise comparisons data are not provided "no statistically significant differences" is not sufficient without inclusion of data in a table or figure.

- Were any drugs listed as submitted but not approved during the study period? Given the temporal aspect of this work, and the uncertain timeline of withdrawals and approvals. More consideration may be warranted with survival-type analysis would be appropriate in some cases, for time to event analysis until approval timeframes (rather than simply assuming these models are not ever approved).

- There was also no discussion regarding consideration of temporal analysis. Either in time to event (i.e. approval) survival-type analyses or adjusted regressions accounting for time period and regulatory changes

=====

Discussion.

The authors touch on many important points. However there remain some disconnections with what was analyzed in this paper. For example, there is no discussion around the differences between "approval vs authorization" and comparisons between countries utilizing each approach.

- The authors also note the addition of specific amendments that may have impacted timelines, as such I would have expected adjusted analysis to account for such.

Reviewer #2: The authors conducted a comparative study on regulatory frameworks utilized under the emergency circumstances, such as the COVID-19 pandemic, in five countries: JP, US, UK, EU, China. One of their claims is that the Chinese government decided to develop drugs for COVID-19 independently. They also showed two more results for the review periods and lags of the specific drugs. The fourth result was the summary of the regulatory frameworks utilized in the pandemic.

In the Discussion, they stated that an emergency approval/use authorization system must be effective in every country. I am not sure what does it mean by “effective” in their terms, and I assumed “effective” means “having a good effect.” It could be very interesting if they specified the characteristics of “effective regulatory frameworks” in emergency situations. This is just a suggestion, but it would be appreciated if they gave some opinions about this point.

Please use CN for the abbreviation of China.

Table 1: “Total” column is confusing, so the column can be omitted. It appears that the Total column is also included in the Chi-squared test.

Table 1: Row name “Withdrawal” is not appropriate, so please consider renaming . I assumed that that category contains drugs that are still yet to be developed

Table 2: Notation of p value is not clear. P< .0001 (the first 0 is missing). In Table 1, there is 0 in front of the decimal point. Please unify the notation.

They stated in Line 154: “The proportion of undeveloped pharmaceuticals in CH was significantly larger compared to other countries (P<0.0001) (Table 2).” It appears that the P value supports the large proportion of the undeveloped pharmaceuticals, which is not the case, I believe. Chi-squared test on Table 2 is merely the difference of the percentage is statistically significant along with countries or development status. Chi-squared test does not imply anything on the specific columns or rows. Please state the precise implication of Chi-squared test with p-value, and then state about specific column and/or row.

In Figure 2, authors compared the review periods in four countries and conducted even statistical test between the US and UK. I believe the comparison is not meaningful because the US already gave permission to use it with the EUA. The situation is somewhat different. My suggestion is to compare three countries JP, EU, and UK, instead.

Line 172 and Line 190: The x-axis represents the average values. -> The x mark represents…

Line 197: Approved drugs (EU: Conditional Marketing Authorisations, JP: Special Approval for Emergency, CH: Conditional Market Approval) can be used… Each item inside of the parenthesis should be corresponding to the word immediately before, “drugs”. Please consider re-phrase like this:

-> Drugs approved with expedited programs (EU: Conditional Marketing Authorisations, JP: Special Approval for Emergency, CH: Conditional Market Approval) can be used even after …

Paragraph: 4. System for approval/usage permission under …

What is the difference among approval, use authorization, and usage permission. I could not follow the discussion.

The first paragraph of the Discussion can be included in the Results section, explaining about Table 3.

6. PLOS authors have the option to publish the peer review history of their article (what does this mean?). If published, this will include your full peer review and any attached files.

Reviewer #1: No

Reviewer #2: No

---

## [Author Response · Author response to Decision Letter 0]

21 Aug 2024

Dear Dr. Ashraful Hoque:

We would like to express my deepest gratitude for your kind consideration. We have diligently revised the Manuscript and related materials to meet PLOS ONE's publication criteria and reviewer's comments. We sincerely hope that our paper will contribute to PLOS ONE and other scientific developments.

Kind regards,

Masahide Kawano

---

## [Editor Report · Acceptance letter]

4 Sep 2024

PONE-D-24-08507R1 

PLOS ONE

Dear Dr. Kawano, 

I'm pleased to inform you that your manuscript has been deemed suitable for publication in PLOS ONE. Congratulations! Your manuscript is now being handed over to our production team.

Kind regards, 

on behalf of

Dr. Ashraful Hoque 

Academic Editor

PLOS ONE